# The Transitional Wettability on Bamboo-Leaf-like Hierarchical-Structured Si Surface Fabricated by Microgrinding

**DOI:** 10.3390/nano12162888

**Published:** 2022-08-22

**Authors:** Ping Li, Jinxin Wang, Jiale Huang, Jianhua Xiang

**Affiliations:** School of Mechanical and Electric Engineering, Guangzhou University, Guangzhou 510006, China

**Keywords:** hierarchical structures, wetting transition, droplet manipulation, microgrinding, Si

## Abstract

Stabilizing the hydrophobic wetting state on a surface is essential in heat transfer and microfluidics. However, most hydrophobic surfaces of Si are primarily achieved through microtexturing with subsequent coating or modification of low surface energy materials. The coatings make the hydrophobic surface unstable and impractical in many industrial applications. In this work, the Si chips’ wettability transitions are yielded from the original hydrophilic state to a stable transitional hydrophobic state by texturing bamboo-leaf-like hierarchical structures (BLHSs) through a diamond grinding wheel with one-step forming. Experiments showed that the contact angles (CAs) on the BLHS surfaces increased to 97° and only reduced by 2% after droplet impacts. This is unmatched by the current texturing surface without modification. Moreover, the droplets can be split up and transferred by the BLHS surfaces with their 100% mass. When the BLHS surfaces are modified by the low surface energy materials’ coating, the hydrophobic BLHS surfaces are upgraded to be superhydrophobic (CA > 135°). More interestingly, the droplet can be completely self-sucked into a hollow micro-tube within 0.1 s without applying external forces. A new wetting model for BLHS surfaces based on the fractal theory is determined by comparing simulated values with the measured static contact angle of the droplets. The successful preparation of the bamboo-leaf-like Si confirmed that transitional wettability surfaces could be achieved by the micromachining of grinding on the hard and brittle materials. Additionally, this may expand the application potential of the key semiconductor material of Si.

## 1. Introduction

Bionic microstructures equipped with manipulable surface properties are in great demand for numerous applications, such as in microfluidic fields [1], optical trapping [2], intelligent sensors [3], etc. To improve the functional properties of the artificially engineered surfaces for broader applications, bio-inspired special wetting interfaces have drawn an ever-increasing amount of attention from scientists [4]. In recent years, the bamboo leaf-like hierarchical-structured surface has stirred growing interest from researchers for its particular wetting property [5,6,7]. For example, based on the bamboo leaf surface’s features, an electronic gas sensor was fabricated to monitor indoor HCHO which required high sensitivity [3]. By bionic manufacturing natural bamboo hierarchical structures, the bamboo/polyaniline triboelectric materials were developed for energy harvesting in extreme environmental conditions [8]. Similar to the rose petal [9], the bamboo leaf also has evolved to possess hierarchical multi-scale microstrips [5], which provide the microstructures that water can stick to, leading to the confined spreading of water even though it contains a great number of hydrophilic groups [10]. The combination of the hierarchical rough structures and hydrophilic chemistry can lead to underwater superoleophobicity [11]. Unlike the superwetting biological surface, bamboo leaves have developed to be hydrophilic but also show the “rose petal effect”, which can maximize the water absorption area and minimize evaporation to meet the needs of their rapid growth [7]. Inspired by this interesting wetting property, researchers have developed advanced functional surfaces based on bamboo hierarchical structures, such as those for anti-condensation [12]. This particular wetting property can also serve as an inspiration for the development of current fluid manipulating systems for water–oil separation [13], mass/heat transfer [14], smart sensors [15], etc.

To further promote developments in related fields, researchers have made great efforts to develop the bio-fabrication of bamboo hierarchical structures [5,6,7,8]. Typical manufacturing processes include using the bamboo leaf as a template to duplicate the multiscale structures on ductile materials, such as polymers [16], metallic materials (copper, aluminum, etc.) [5,7,17]. However, the template-free, eco-efficient fabrication of hierarchical structures on hard and brittle materials is still challenging since the manufacturing mechanism differs from those of ductile metals. As a typical hard and brittle material, monocrystalline silicon (Si) is usually used in microelectronic chips [18], biosensing [19], molds, etc. The machining of a micro/nanostructured Si surface without a crack is extremely difficult from the standpoint of the removing mechanism [20]. The surface structuring methods based on high energy density beams have appeared as state-of-the-art fabrication techniques for hard and brittle materials, including e-beam lithography [21], femtosecond laser structuring [1], and two-beam laser interference patterning [22]. Although the beam-based methods have made great contributions to scientific and social development, they are inseparable from high manufacturing costs, low removal efficiency, and limited accuracy controllability [18]. Moreover, these manufacturing methods were usually accompanied by subsequent coatings or modifications of low surface energy materials [1,22] to transform the surface wetting property. Nevertheless, surface coatings are unstable at high temperatures and in extreme environments, such as the boiling, condensing, or in acid/alkali. Therefore, it is still a great challenge to develop a transitional wettability surface with environmental stability along with an eco-efficient technological process.

With the continuous attention on sustainable development, ultraprecision cutting using artificial diamond tools has emerged as an eco-friendly and cost-effective fabrication method for micro-structuring functional surfaces on hard and brittle materials [23]. Diamond cutting tools can generate micro/nanostructures to several tens of nanometers owing to the extreme sharpness of diamond grains [24]. Two typical processes based on diamond cutting have been proven for the controllable and flexible manufacturing of microstructured functional surfaces, namely, shaped grinding-wheel patterning [25] and fly cutting [26]. Fly cutting produces microstructures by directly replicating the shape of only one grain on the tool tip. The main drawback is that the generation rate of fly cutting is very low (typically only 50 Hz) even though the spindle speed reaches 3000 rpm [27]. In contrast, shaped grinding-wheel patterning utilizes sharpened grains on the dressing grinding-wheel to patter microstructures down to the submicron scale through a large number of diamond grains periodically interacting with the material [28]. By structuring the grinding-wheel’s working surface and programming its working path, the workpiece can be machined to various microstructure arrays [29]. However, due to the hard and brittle material easily generating brittle fractures resulting in a failure of the bionic functional surface, most of the available outcomes of ultraprecision cutting for single-scale structural surfaces were achieved [30]. The microginding fabrication of multi-scale hierarchical microstructures, especially the bio-inspired functional surfaces, is still challenging and unproven. Till now, a simple way of microgrinding to realize the bionic wetting functional Si surface has yet to be found in the published literature.

In this paper, the bamboo-leaf-like hierarchical-structured (BLHS) surfaces were fabricated on the Si chips by using a diamond grinding wheel with one-step forming. This special structure transformed the wetting property of Si from hydrophilic to hydrophobic. This is unmatched by current wetting modification surfaces, which rely on both micro-structuring and coating. The surface transitional wettability and wetting robustness of the BLHS surfaces were investigated by combining fractal feature characterization and droplet manipulating behaviors. Stochastic process methods, such as a correlation function and the Abbott–Firestone curve, were introduced to characterize the BLHS morphologies. Additionally, the roughness factors and 3D box-counting methods were introduced to extract the key morphological parameters of the hierarchical multi-scale structures. The fractal wetting model was proposed for numerically studying the relation between BLHS surface features and their wetting properties in a stable transitional wetting state.

## 2. Experiments

### 2.1. Structural Characteristics of Bamboo Leaf for Droplet Steering

Natural bamboo is endowed with the ability to assemble different hierarchical structures on varisized scales, and these hierarchical structures result in bamboo’s superior macroscopic properties [31]. The bamboo leaf is found to consist of hierarchically arranged heterogeneous micro-grooves (micro-strips) toward the leaf vein (Figure 1a,b). The depth of the grooves mainly ranges from 0.1 to 10 μm for mature bamboo leaves (see Figure 1c) from the Abbott–Firestone curve of the bamboo leaf morphology. For seedling leaves, the depth of the grooves is even smaller, mainly ranging from 0.05 to 5 μm. The bamboo leaf with nano-micro grooves exhibits hierarchical topology. The water contact angles on the front side and back side of the bamboo leaf are ~82°and ~135° (Figure 1d,e), respectively. The surfaces exhibit typical hydrophilic and hydrophobic properties, respectively. Moreover, the droplet always sticks to the leaf and keeps the coronal shape even when the leaf is placed vertically (Figure 1f). The hydrophilic feature on the front side of the leaf can maximize the liquid contact area as well as the liquid contact time to keep it fresh. The wetting difference allows the leaves to stretch to the photosynthetic rate in the hydration state and curl up to reduce evaporation in the hydropenia state (Figure 1g,h).

### 2.2. Microfabrication of Bamboo-Leaf-like Hierarchical-Structured (BLHS) Surfaces

Figure 2 is a schematic diagram of bamboo-leaf-like hierarchical microstructures micromachined by a diamond grinding wheel on a Si surface. The diamond grains were randomly distributed on the grinding wheel edge with various grain protrusion heights and volumes (Figure 2a). When the diamond grinding wheel microground in the plastic zone along a given direction, the microgrooves were randomly and hierarchically patterned on the Si surface. In this study, diamond grinding plane wheels with diamond grain particle sizes from SD60 to SD3000 (Appendix A gives a particle size conversion chart) were used to fabricate micro–nanometer grooves on Si surfaces with the same micro-grinding conditions (see Table 1). Given a grinding wheel speed *N* and feed speed *v_f_*, the diamond grains gradually cut into and cut out the Si surface along the cycloid track (Figure 2c). When micro-cutting is performed in elastic-plastic removal region, the microgrooves are gradually patterned by successively replicating diamond grain profiles on the workpiece’s surface. The top edges of the grains cut the material surface during microgrinding; thus, the angle of a microgroove was determined mainly by the top angle of the micrograin (Figure 2b). Generally, the top angles of the diamond micrograins were approximately 90° to 170°.

## 3. Materials and Measurement

The Pink single bamboo (also called Bambusa chungii McClure) was collected from Guangzhou, Guangdong, China. The monocrystalline silicon was purchased from Senshuo Technology Co., Ltd., Zhejiang, China. Before testing the surface wettability, all the samples were ultrasonically washed in absolute alcohol and thoroughly rinsed in running deionized water. The droplet contact angles were measured using a Dataphysics OCA40 Micro (Filderstadt, Germany). The volume of the DI water droplets used was 3 microliters. Surface wettabilities were tested at a constant temperature of 25 °C and relative humidity of 58%. All the data were determined by averaging five individual measurements. The three-dimensional (3D) topographies of the micro-machined BLHS surfaces are measured by a white light interferometer (WLI: BMT SMS Expert 3D). The Abbott–Firestone curve parameters were computed using a Taly Map Platinum measurement software available as a part of BMT SMS Expert 3D. The two-dimensional (2D) topographies were completed by scanning electron microscopy (SEM: TESCAN MIRA4). The SEM photos of fresh bamboo leaves were measured by an environmental scanning electron microscope (ESEM, FEI QUANTA250). The optical observations were performed by a superdepth microscope (Keyence VHX-2000). To accurately character the morphology and the wettability, more than 100 bamboo leaves and the BLHS surfaces fabricated by the diamond grindings of SD60~SD3000 were measured.

## 4. Results and Discussion

Figure 3 shows the SEM photos of the BLHS morphologies with the microstructures in nanometer to micron depths. In diamond grinding, a Si surface is randomly cut by diamond abrasive particles distributed on the working surface of the grinding wheel’s periphery. The textures on the microground surfaces were similar to the bamboo-leaf surface in terms of the microscopic morphology. Although the directions of the microgrooves were parallel with the mcirogrinding path, the sizes, shapes, and depths of the microgrooves with random nanostructures that appeared on the micromachined surface were different (see enlargements in Figure 3), which shows the hierarchical scale character. Additionally, coarse diamond grains, such as the SD60-SD 320 diamond grinding wheels, were more likely to cut deeper (Figure 3b), and finer diamond grains (such as SD 3000) were more likely to pattern multiscale grooves (Figure 3d) under the same load and wheel speed. That was because a decrease in grain size brought an increase in the number of grains and abrasive particles for the same grinding wheel dimensions (wheel diameter, width, etc.). Unlike the lapping [32] and polishing [33] process methods in which the abrasive grains undergo directionless movement during the machining process, all the diamond grains on the grinding wheel tip’s surface are cutting the surface along the same cycloid track (see Figure 2), resulting in the consistent direction of the microgrooves.

Due to the hierarchical features of structures on the microgrinding surface, it is necessary to characterize the difference in surface roughness in different directions. The arithmetic mean value of the surface profile along the microgrinding trajectory is defined as the roughness *Ra_x_*, the arithmetic mean value of the profile normal to the microgrinding direction is the roughness *Ra_z_*, and the arithmetic average deviation value of the mapping solid area is defined as the solid-surface roughness *Sa*. A schematic of the surface roughness is shown in Figure 4a. Figure 4b shows the measured values of the profile along and vertical to the microgrinding trajectory. It indicates that the *Ra_x_* apparently differed from the *Ra_z_*, showing a strong roughness anisotropy. It is easily reasoned that *Ra_z_* was always larger than *Ra_x_* because the microstructures were mainly perpendicular to the grinding direction. In contrast, the nanostructures were distributed primarily along the grinding trajectory. Table 2 gives the solid–surface roughness values of each sample surface. The solid-surface roughnesses of bamboo leaves covered the entire lifecycle. The results indicate that the solid–surface roughness *Sa* increases with diamond grain size.

To effectively extract the morphology features of hierarchical microstructured surfaces, a stochastic process method, the Abbott–Firestone curve, was used to overcome the shortcomings of average roughness, such as the roughness *Ra* or *Rz.* The Abbott–Firestone curve uses parallel lines to show the proportion of the data in a whole surface contour data point cloud to describe the surface texture properties [34] and represents the probability density function of surface height contours [35]. Figure 5 shows the Abbott–Firestone curves of the morphology of the BLHS surfaces obtained by microgrinding. The curves were derived by applying the correlation function of statistical methodology to characterize these surfaces. The histogram in the figure gives the proportion for a given depth of microstructure. The curve marked in red shows the microstructured surface area ratio as a function of depth. The figures show that both the natural bamboo leaf (Figure 1c) and the BLHS surface matched the asymmetric generalized Gaussian distribution. The depth of the nanostructures mainly ranges from 50 to 100 nm, and the depths of the microstructures mainly range from 0.1 to 5 μm. The coarse diamond grinding wheels (SD 60 and SD 320) produced the microstructures with larger depths primarily ranging from 2 to 3.5 μm, and the fine diamond grinding wheel (SD 3000) created the microstructures with relatively smaller depths ranging mainly from 0.5 to 0.7 μm. The Abbott–Firestone curves of the surface morphologies confirmed that microgrinding can bio-fabricate the bamboo-leaf–like surfaces with hierarchical characteristics on the hard and brittle material of Si.

Although a statistical methodology can characterize a hierarchical-structured surface to some extent, for engineering applications, the information sometimes must be perceived directly. From the Abbott–Firestone curve of the surface morphology, it was determined that the BLHS surfaces met the fractal requirements of complexity and multi–scalarity. It could be clearly ruled out that the fractal theory was a suitable choice to characterize the complex details of a BLHS surface. The box-counting method [36,37,38] is generally used to calculate the fractal dimension *D_f_* of a fractal surface because it is easy to do by tracing the curve of the surface cross-section. However, that is a one-dimensional (1D) calculation method. To obtain a more accurate numerical value, the 2D fractal dimension *D* is proposed to calculate the actual fractal dimension of the 3D surface.

However, the 2D fractal dimension *D* proposed by calculating *D* ≅ *D_f_* + 1 on the base of the box–counting method [36] is actually a 1D fractal dimension from another point of view. Jiang et al. [37] calculated the 2D fractal dimension *D* by using image identification programming to extract the texture of a scanning electron microscope image. However, some details of the surface profile features could be easily overlooked when viewing the image. In this paper, a straightforward and accurate characterization method is proposed to quantitatively analyze the 2D fractal dimension of a hierarchical-structured surface by using a 3D box-counting method to characterize the solid surface directly. In accordance with the 3D box-counting method, the fractal dimension *D* of an irregularly microtextured surface can be described by the following equation:(1)$$D=\lim(-\frac{\ln N_{\varepsilon }}{\ln\varepsilon })$$
where *ɛ* is the length of box, and *N**_ɛ_* is the number of the boxes.

For a fractal surface, its fractal features are determined by the size of the box. When the box is too small or large, the fractal features of the surface are concealed. Therefore, the fractal features are valid only within a certain scale. Consequently, a fractal scaleless band was used to solve the distortion of the fractal function:ln*N_ɛ_* = −*D*ln*ɛ* + ln*C*(2)
where *C* is the intercept of the double log plot of Equation (1), and it is usually a constant. The scale-free interval is determined by the slope of Equation (2). This method can effectively find the linear part of the function and obtain the accurate fractal dimension. The fractal dimension *D_β_*, which characterizes the self-similarity of micromachined irregular surfaces, can be derived from the following equation [38]:(3)$$N\left(l_{\varepsilon }\right)\propto \pi^{-1}Sl^{-D_{\beta }}$$
where *l* is the box size, *N*(*l**_ɛ_*) is the number of boxes, and *S* is the covered area of the surface. The fractal dimension *D_β_* is obtained from the slope of the function ln*N*(*l**_ɛ_*)/ln*l**_ɛ_*.

The fractal dimensions of the measured surfaces in this study are shown in Figure 6. The 3D surface information (see the inset image) was directly obtained by the white light interferometer measurement data of the surface. The fractal dimensions of the natural bamboo leaf are between 2.10 and 2.61, while the BLHS surfaces are between 2.06 and 2.54. This coincides with the conclusions of references [37,38] about the fractal dimension (2 ≤ *D* < 3) of a solid fractal surface. The fractal dimensions of the microground surfaces decrease with the increase in diamond grain size, which is contrary to the inclusion of solid surface *Sa*. Two main reasons are involved: firstly, the fractal surface with more scaled structures will produce a larger fractal dimension value; secondly, in the calculation of surface roughness, all the measurement points are averaged, and the multidimensional and multiscale structures are often overlooked. This further illustrates that the hierarchical structures patterned by fine diamond grinding result in an apparent multiscale effect. The fractal dimensions of each sample surface have been shown in Table 2.

Figure 7 shows a visible difference in contact angles on the micromachined surfaces. It showed the wetting property transition from a hydrophilic state to a hydrophobic state using only surface microstructuring by various types of diamond grinding wheels. The bare Si was intrinsically hydrophilic (with a contact angle less than 60°), as seen in Figure 7a. The contact angles increased from the bare Si surface to the microground Si surfaces as the diamond grain sizes increased. The BLHS surfaces micromachined by the diamond grinding wheel with grains larger than SD320 were hydrophobic, with contact angles ranging from 92° to 97° (Figure 7b,c). It is worth noting that the droplets on the microground Si surfaces all had spherical coronal shapes. This shape was caused by the droplets being dipped into the gaps formed between microstructures, where the fringes of the microstructures prevented the droplets from spreading to larger areas. Thus, the droplets formed more spherical crowns, leading to increases in the contact angles. The contact angle hysteresis measurement (Figure 7f) showed that the droplets clung to the microground Si surfaces and remained spherical regardless of the tilt angle; this was, however, consistent with the front surface of the bamboo leaf (Figure 1f). The above results indicate that microgrinding can bio-fabricate the bamboo-leaf-like surface both in terms of morphology and wetting performance.

To assess the wettability mechanism of those BLHS surfaces, the three-phase contact line was inspected and investigated. The details of the contact situation between a droplet and the BLHS surface are shown in Figure 8. At the solid–liquid contact interface, the liquid filled the space among the microstructures, and no bubbles or air pockets were trapped from the micrometer scale observation (Figure 8a,b). Different wetting phenomena were shown along the regular circular contour of a droplet: parts of the droplet parallel to the *y*-axis (marked by blue arrows in the figure) sank into the bottom of the microstructures, whereas some parts parallel to the *x*-axis (marked by red arrows) were pinned at the tops of the microstructure ridges. This indicates that the Y-partial droplet was under total wetting, and the X-partial was in a local metastable state. Enlarging the solid–liquid–gas contact part by 300, 500, and 1000 times, respectively, showed that the three-phase contact line was continuous, and no “sawtooth” was found along the droplet contour. This indicates that the liquid impregnated the microgrooves (Figure 8d,f), and from a visual inspection, the droplet was not in a Cassie wetting contact. However, because both the contact angles and adhesion are large, there will remain air-chamber patterning among the “sinking” liquid and nanostructures (Figure 8c). Because the BLHS surface owns the hierarchical micro- and nanostructures, it is suggested that the wetting regime of the droplet is in the transitional wetting state rather than the totally wet contact mode.

To explore the hydrophobic wetting stability and to further verify the wetting state of the BLHS surfaces, a droplet impact experiment was conducted. Figure 9 shows the results of the experiments. The droplet impact experiment is a widely used method to test the wetting stability of a surface by evaluating the ratio of kinetic energy to surface energy—the Weber number *We* [30,39]. When the droplet is released from a given height, it hits the surface with a velocity *v*. *We* is a function of impact velocity *v*. The numbers in Figure 9 represent the droplet states before, during, and after impact: Serial number 1 is before impact, 2–7 are during impact, and 8 is after impact. The droplet shape and state on the BLHS surface were maintained before and after the impact, and the contact angle barely decreased. This suggests that the hydrophobic wetting state is relatively stable on the BLHS surface. As the impact velocity has reached 1.0 m/s, the change from the original intrinsic hydrophilic surface to the hydrophobic surface is not caused by the microstructures trapping the air. In addition, the contact angle was reduced by only 2° after the droplet impact, which suggests that the air pockets trapped in the nanostructures are firmer compared to the microstructures. The results further confirmed the conclusions in Figure 8 that the droplet was in a stable transitional wetting state on the BLHS surface.

Inspired by the wetting difference between the front surface and back surface of bamboo leaf, we modified the BLHS surface to be hydrophobic by surface coating with the low surface energy material of photocurable perfluoropolyether (PFPE) by UV illumination photocuring. Then the droplet steering ability was tested. The water droplet can be split up equally when used with two pieces of BLHS surface (Figure 10a and Appendix A). When the BLHS surface is coated with PFPE, the water droplet can be totally taken away by the BLHS surface (Figure 10b and Appendix A). The reason for this difference is that the surface energy of the modified BLHS surface is much lower than the directly micro-machined BLHS surface, which can be further proved by the droplet manipulation tests in Figure 10c,d and adhesive strength test in Figure 11. The droplet was running with the needle without volume loss (Figure 10c and Appendix A). This indicates that the droplet on the modified BLHS surface is in nonwetting state (Cassie’s state) with low contact angle hysteresis [10]. The wetting property of modified surface is similar to the one of the back side of the bamboo leaf, as mentioned before. When replacing a needle with a springe needle, the droplet completely self-sucked into the needle tubing within 0.1 s without the help of external forces (Figure 10d and Appendix A). As opposed to what had been observed on the modified BLHS surface, the droplet stuck on the BLHS surface concerning the needle droplet manipulation (Appendix A). These observations suggested that the droplet on the BLHS surface can be manipulated, with a high adhesive surface preferring droplet separation, whereas a low adhesive surface preferring droplet transferring.

To explain the mechanisms responsible for the droplet adhesion property on the BLHS surfaces, the surface adhesive ability was examined. We designed an experimental device to observe droplet morphology in the state of imminent desorption and collapse through tensile and compression tests (Figure 11). The adhesive strength is characterized by tensile and compression ratio. The tensile (compressive) ratio is defined as the absolute difference value of tensile (compressive) height and original height to the original height Δ*h* = |*h* − *h_0_*|/*h_0_*. When using a BLHS surface to pull a droplet (3 μL) sitting on the same BLHS surface (Figure 11a), the tensile height *h* is 1.79 mm, which is close to the spherical droplet diameter. When using a BLHS surface to push the droplet sitting on the BLHS surface (Figure 11c), the compression height *h* is 0.447 mm. Significantly, the tensile ratio is about 40.8% and the compression ratio is about 66.6%. This indirectly proves that the adhesion force is greater than the gravity of the droplet itself. Additionally, this indicates that it is easier to compress the droplet than to stretch it if used on the same surface. It is because the droplet has to overcome gravity while stretching up. Additionally, we used two different surfaces to conduct the experiments. The droplet on the BLHS surface can also be stretched up and compressed by using a flat-end needle, but it is easier to stretch up rather than to squeeze (Figure 11b,d). This is because the needle will prick the droplet during compression. If we use the modified BLHS surface to stretch up the droplet, however, the droplet is unmoved and not attached to the modified surface.

To explain the mechanisms responsible for the microground BLHS Si surfaces exhibiting a transitional wetting property with a large contact angle and surface adhesion, we resorted to the force analysis to explore how a droplet is confined to its spreading wetting (Figure 12). Regardless of the liquid gravity, three main forces act on the droplet: the wicking force *F_c_*, the surface tension *τ,* and the pinning force *F_p_*. The wicking force *F_c_* is produced by the re-entrant microgrooves, and it causes a downward force on the droplet. The liquid surface tension *τ* produces an opposite resistance to lift the droplet. Both the wicking force and surface tension have the component forces to restrain the droplet from spreading out. In addition, the fringes of the microgrooves pin the droplet and produce the corresponding pinning force *F_p_*, preventing the liquid from completely wetting the surface. Thus, the wicking combined with the re-entrant effects result in an increased contact angle and a transition in the wetting property.

Wenzel and Cassie–Baxter wetting models [40,41] are commonly used to describe the value of the apparent contact angle of a droplet on a textured surface for the wet-contact mode and nonwet–contact mode [42], respectively. Figure 13a,b show the relations between contact angles and roughnesses on hierarchical microstructured surfaces predicted by classic wetting models. The green area in the figure represents the general range of the measured contact angles on the bamboo surfaces. According to the Wenzel and Cassie–Baxter models, the contact angles decreased with the increase in Wenzel roughness factor *γ_w_* (Figure 13a) and Cassie roughness factor *γ_c_* (Figure 13b), which are respectively defined as the ratio of actual contact area to the apparent area and the fraction of the droplet’s base area contacting with the solid. The contact angle was always less than 90° for an intrinsic hydrophilic surface, as predicted by the Wenzel model. However, the experimental results showed contact angles larger than those predicted by the Wenzel model but much smaller than the Cassie–Baxter predicted values. There are three reasons for this: firstly, the Wenzel state represents a wet-contact mode of water on a rough surface and the Cassie state represents a non-wet-contact mode; secondly, the single-variable measurement of the hierarchical surface morphology results in the difficulties of quantitative characterization of actual surface features [35]; thirdly, there is a stable intermediate state, as depicted in Figure 8c, in which the liquid sinks in the microstructure but the air-chamber patterns among the “sinking” liquid and nanostructures. This can be verified by the fact that the trend of measured contact angles on the modified BLHS is in accordance with the Cassie–Baxter curve as shown in the Figure 13b, marked by red circles.

Because the hierarchical structures in the micromachined surface had obvious fractural features, the fractal theory was applied to characterize the wetting for the BLHS surfaces. The wetting state of a droplet was in a noncomposite mode, as previously shown; therefore, a transformation of the Wenzel equation was performed based on the fractal theory to suit it to hierarchical structures. Combined with Equation (3), the Wenzel equation can be transformed as Equation (4). A similar transformation was also performed in Refs [36,37].
(4)$$\cos\theta =(D_{\beta }-D)r_{s}\cos\theta_{e}$$
where *θ* is the calculated contact angle, *D_β_* and *D* are the fractal dimensions of the micromachined surfaces and the fractal dimension of the intrinsic surface, and *θ_e_* is the intrinsic contact angle. Here, the *D* of the intrinsic surface was 2.06, the intrinsic contact angle was set as 57°, *r_s_* was the ratio of the actual contact area of the droplet to the projected area. The parameters are determined according to the apparent wetting features of the micromachined surface.

Figure 13c shows the relationship between the measured contact angles and the fractal wetting model. The green area in the figure is the general range of the measured contact angles on the front side of the bamboo surfaces. It shows that the contact angles decrease with the increase in fractal dimensions. Both trends of the measured contact angles on the BLHS surfaces and the contact angles on the bamboo leaves agree well with the values calculated by the proposed fractal wetting model. This is because the equilibrium contact angle on the textured surface is not only governed by the liquid tension but also governed by the resultant dynamic force during droplet spreading [43]. Compared to the fractal wetting models proposed by Onda [36] and Jain [44], which are used in the condition of Cassie’s superhydrophobic wetting state, the proposed fractal wetting model is a good supplement to the noncomposite wetting state.

The wetting state of a surface is usually a combination of its chemical and physical properties. Energy-dispersive X-ray spectroscopy (EDS: TESCAN MIRA4) was performed on the microground surface (the measured area shown in Figure 14a). The EDS image confirmed that Si was the only element presenting on the surface after microgrinding (Figure 14b). It indicated that the microgrinding process can microfabricate the bamboo–leaf–like hierarchical structures on the single crystal silicon surface without changing the material’s chemical properties. Although there were small traces of native oxide, they were also present on the original silicon surface. The EDS results demonstrate that the wetting transition of the BLHS silicon surface was caused by the hierarchical structures rather than chemical surface modifications.

## 5. Conclusions

The BLHS surfaces can micromachine by diamond grinding with one-step forming, obtaining a stable transitional wetting property without surface chemical modification or coating. Results show that the BLHS surfaces not only showed fractal characteristics but also yielded a turned wettability. Key parameters—such as solid-surface roughness, the Abbott–Firestone curve, and fractal dimension—were proposed to characterize the morphologies of the BLHS surfaces with respect to the roughness factor and surface roughness. The obtained BLHS surfaces are not only similar to bamboo leaves in morphology but also in wetting properties. The fractal dimensions of the micromachined BLHS surface are in the range of 2.06~2.54, and the wetting angles are in the range of 77° to 97°. Additionally, the contact angle was only reduced by 2% after the droplet impacts. The fractal dimensions of the micromachined surfaces decrease with increase in diamond grain size while the solid surface roughness increases with the increase in diamond grain size. The fractal dimension is more effective in characterizing the morphology features of the BLHS surface than the surface roughness and the roughness factor. Based on the fractal characteristics of the BLHS Si surface, a fractal wetting model was used to study the numerical relation between the morphology of the hierarchical structure and its transitional wettability. The results demonstrate that the fractal wetting model can be used to explain the wetting properties of BLHS surfaces, indicating that the fractal microstructure has a significant influence on the surface wetting properties.

## Figures and Tables

**Figure 1 nanomaterials-12-02888-f001:**
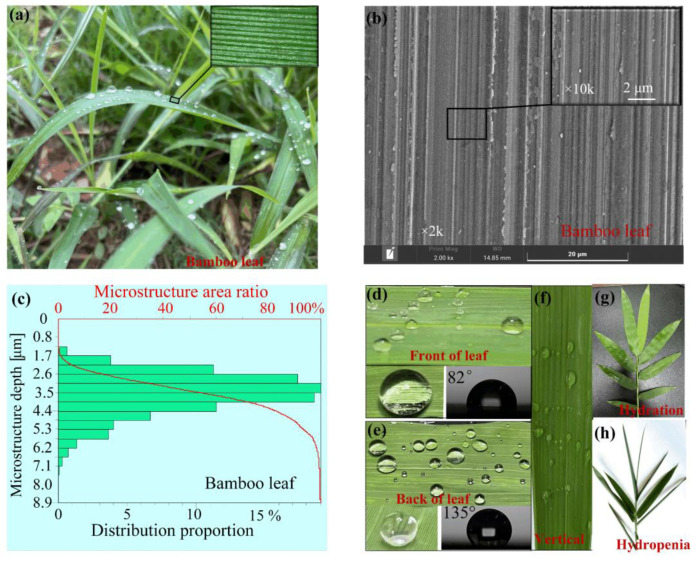
Structural characteristics of bamboo leaf for droplet steering. (**a**) The morphologies of natural bamboo leaf; (**b**) the SEM photo of the fresh bamboo leaf using an environmental scanning electron microscope; (**c**) the Abbott–Firestone curve of the morphology; (**d**,**e**) are photos of bamboo leaf in the hydration and hydropenia states, respectively; (**f**) droplets pinned on the vertically placed leaf; (**g**) and (**h**) are wetting states on the front side and back side of the leaf, respectively.

**Figure 2 nanomaterials-12-02888-f002:**
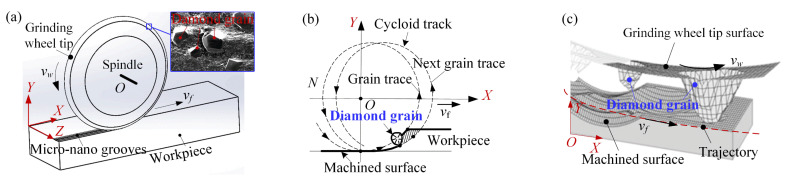
Schematic diagram of BLHS surface microground by a diamond grinding wheel. (**a**) The grinding of micro-nanogrooves; (**b**) the micro-grinding mechanism of the diamond grain; (**c**) the removal process of the involved micrograins on the grinding wheel tip surface.

**Figure 3 nanomaterials-12-02888-f003:**
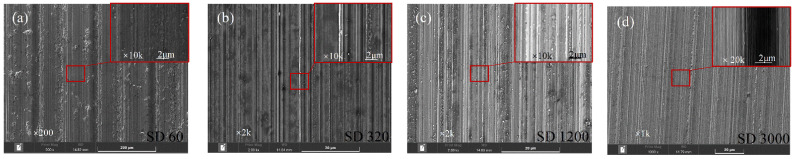
Morphologies of the BLHS Si surfaces fabricated by (**a**) SD 60, (**b**) SD320, (**c**) SD1200, and (**d**) SD3000 diamond grinding wheels.

**Figure 4 nanomaterials-12-02888-f004:**
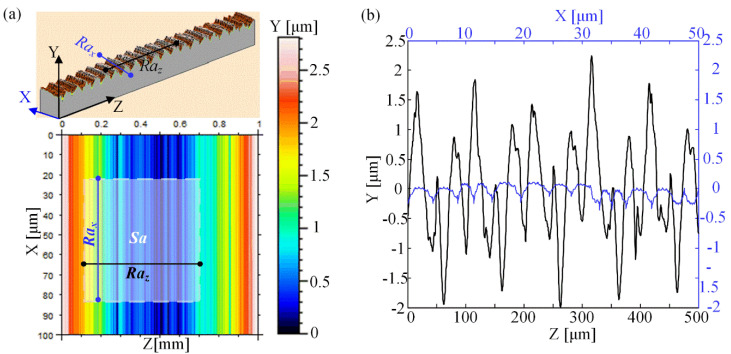
Roughness of Si surface. (**a**) Schematic illustration of roughnesses *Ra_x_*, *Ra_z_*, and *Sa*. (**b**) Profiles of the BLHS surface.

**Figure 5 nanomaterials-12-02888-f005:**
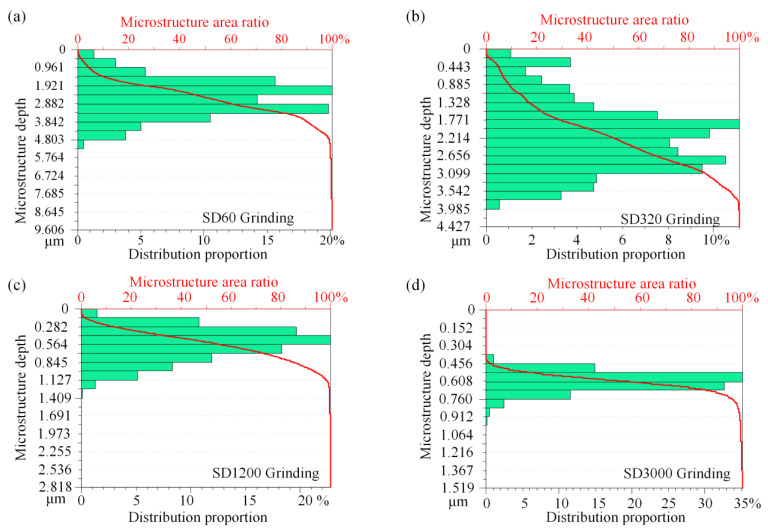
Characterization of BLHS surfaces by Abbott–Firestone curves. Surfaces micromachined by (**a**) SD60, (**b**) SD320, (**c**) SD 1200, and (**d**) SD 3000 diamond grinding wheels, respectively.

**Figure 6 nanomaterials-12-02888-f006:**
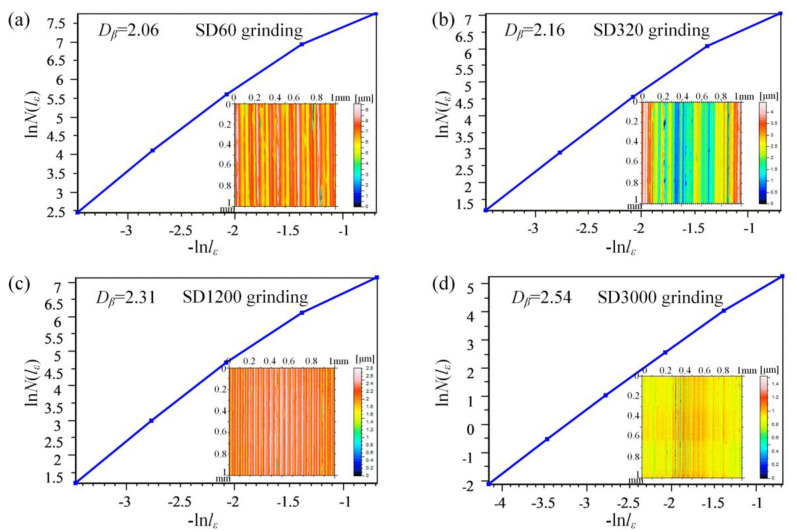
The fractal dimensions of the micromachined BLHS surfaces. The subgraphs are the measured surface topographies.

**Figure 7 nanomaterials-12-02888-f007:**
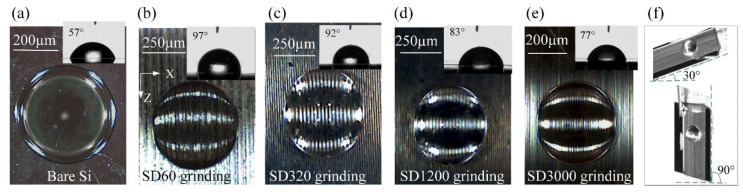
Static wettability states of droplets on the microfabricated BLHS surfaces. The subgraphs are the contact angles of the measured surfaces.

**Figure 8 nanomaterials-12-02888-f008:**
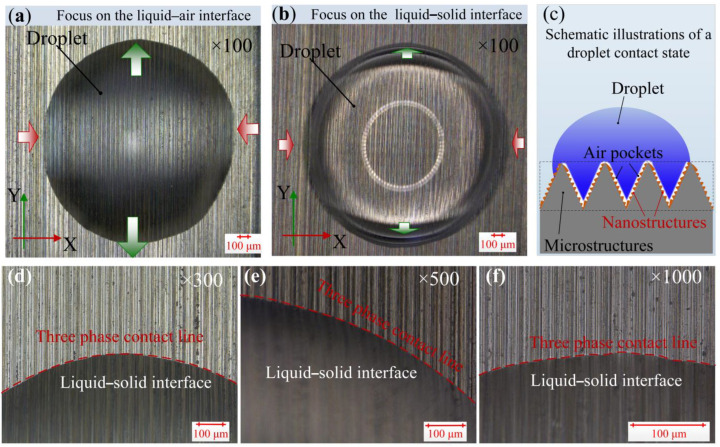
Digital microscopic images of droplet contact on the BLSH surfaces. (**a**,**b**) Contact morphologies of the droplet; (**c**) schematic illustration of the droplet contact mode; (**d**–**f**) enlarged partial views of contact at various magnifications.

**Figure 9 nanomaterials-12-02888-f009:**
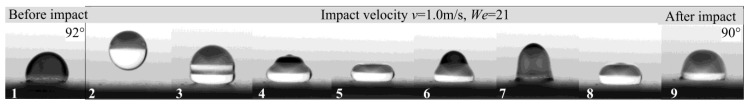
Droplet impact state of the microground BLHS surface.

**Figure 10 nanomaterials-12-02888-f010:**
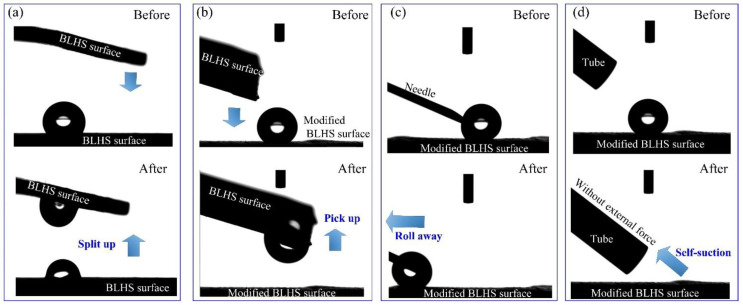
Droplet manipulation tests on (**a**) BLHS surface by a BLHS surface; (**b**) modified BLHS surface by a BLHS surface; (**c**) modified BLHS by a needle; and (**d**) modified BLHS surface by a tube.

**Figure 11 nanomaterials-12-02888-f011:**
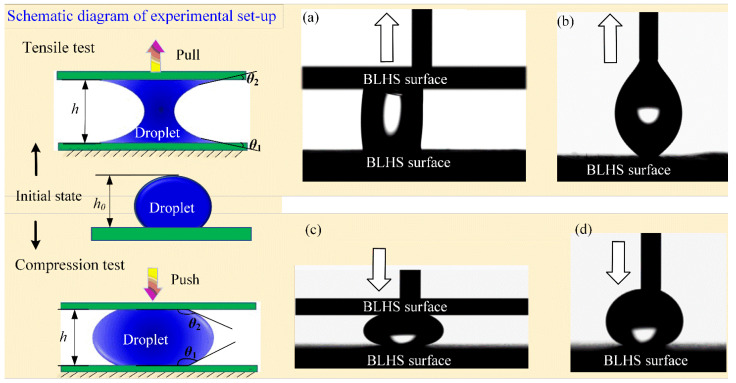
The BLHS surface adhesive ability experiments. Droplet (**a**) stretched up by a BLHS surface; (**b**) stretched up by a flat-end needle; (**c**) compressed by a BLHS surface; and (**d**) compressed by a flat-end needle.

**Figure 12 nanomaterials-12-02888-f012:**
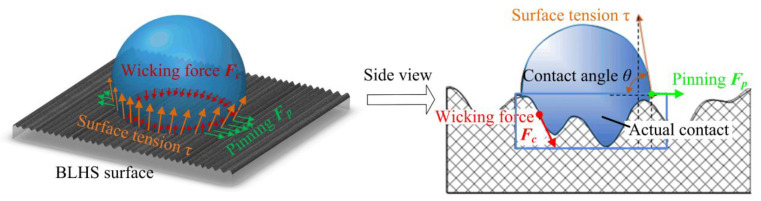
Force analysis of the droplet on the BLHS surface.

**Figure 13 nanomaterials-12-02888-f013:**
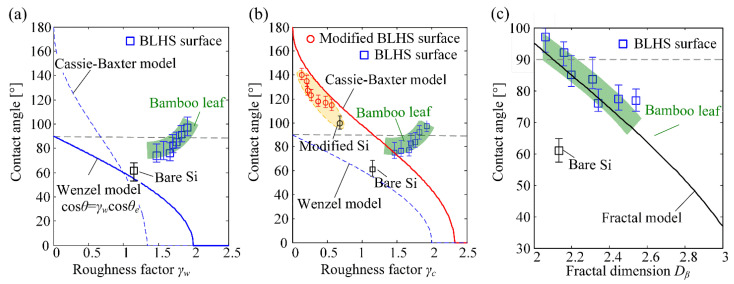
Comparison of the measured contact angles and theoretical values calculated from various wetting models. (**a**) Wenzel wetting model, (**b**) Cassie wetting model, (**c**) fractal wetting model.

**Figure 14 nanomaterials-12-02888-f014:**
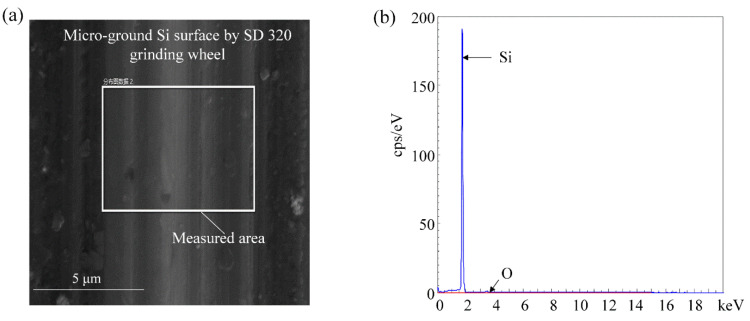
(**a**,**b**)EDS measurement of the microground surface.

**Table 1 nanomaterials-12-02888-t001:** Microgrinding conditions.

CNC Grinder	SMART B818
Diamond grinding wheel	SD 60~SD3000, Resin bonded, *N* = 3000 rpm, *v*_f_ = 2000 mm/min
Tool paths	Level, linear
Workpiece	Monocrystalline silicon
Fine grinding	X-direction: *a* = 1.5 µm
Coolant	Water

**Table 2 nanomaterials-12-02888-t002:** Solid surface roughnesses and fractal dimensions of the sample surfaces by micromachining with different diamond grindings.

Sample	SD60	SD120	SD320	SD600	SD1200	SD2000	SD3000	Bamboo Leaf
*Sa* (μm)	7.2	6.3	4.6	4.0	1.2	1.0	0.6	0.7~8.6
*D_β_*	2.06	2.13	2.16	2.21	2.31	2.42	2.54	2.1~2.61

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
