# Peer review of "The Transitional Wettability on Bamboo-Leaf-like Hierarchical-Structured Si Surface Fabricated by Microgrinding"

_nanomaterials, 2022, doi:10.3390/nano12162888_

Round 1

Reviewer 1 Report

Manuscript ID: nanomaterials-1851818 entitled:

The transitional wettability on bamboo-leaf-like hierarchical-structured Si surface fabricated by microgrinding

Authors: Ping Li , Jinxin Wang , Jiale Huang , Jianhua Xiang

General comment

The authors present the results regarding the possibility of controlling the contact angle, by adjusting the geometric parameters of the surfaces. The fabrication of artificial materials with hydrophobic and superhydrophobic properties that mimic the morphologies of plant leaves is promising. Thus, the successful preparation of Si as bamboo leaves confirmed the feasibility that transition wettability surfaces can be fabricated by micromachining.

Some recommendations and observation remain:

1. Please include specific and quantitative results in abstract, in a manner suitable for a broad audience. Emphasis the novelty of the work in Introduction section and correlate with the state of art about this subject reported in the literature. 

2. In Fig 1c the Abbott–Firestone curve of the morphology, please insert the titles for all axes and link the curve to the proper Y axe (or change the colour of Abbott–Firestone curve in red). Also, for axes in fig 5. Explain in text the complete meaning of figures. For example, the first graph displayed in figure 5 as a Histogram, gives the probability (frequency) for points to be at a given height (depth). The second graph Abbott-Firestone curve, is the cumulative curve of the distribution, displays the material ratio in function of the depth. The material ratio is the percentage of the surface above a given depth.

3. Describe all procedures by which the surface profile data were obtained, analyzed (software ……). Please, provide complete information about the fractal description used in the study. Each term in the presented equations must be explained.

4. Specify, the specie of investigated bamboo. Briefly describe the methodology follow in order to   minimize dehydration during transportation to nearby laboratory facilities (preserved in water?) How were performed SEM analysis (on dried/wet pieces of leaves)?

5. At presented magnification of leaf surface (figure 1b), some microstructure projections covering all the surfaces (micropapillae) and wax bumps (at nano-scales) are usually observed. Can author comment this aspect?

6. It is known that the surrounding viscous medium or the size of the surfaces relative to the droplets could have influenced the wetting of the droplet by introducing edge effects. The proposed model can be used as such in the case of other substrates or viscous media or does the present study need to be accommodate?

7. Insert the correlation factor between preparation technologies and D. It was noted that the trend of D value of the fracture surfaces was lower when Sa was high, the dependence is almost linearly (R2~0.87).

Figure 13c shows the relationship between the measured contact angles and the fractal wetting pattern. The author writes: "The green area represents the contact angles measured on the front side of natural bamboo leaves in different growth phases" Explain what contact angles mean in different growth phases.

8. Insert in conclusions also a short sentences with the results obtained for solid surface roughnesses and fractal dimensions of the sample surfaces by micromachining with different diamond grindings. Please include specific and quantitative summary results in your Conclusions suitable for a broad audience.

9. Revised the English

For example contact angles instead of  contact angels.

Author Response

  1. Please include specific and quantitative results in abstract, in a manner suitable for a broad audience. Emphasis the novelty of the work in Introduction section and correlate with the state of art about this subject reported in the literature.

Response: The abstract has been revied and specific and quantitative results have been added.

The Introduction has been revised and the novelty of the work has been highlighted in Introduction.

The novelty of the work is: The machining of micro/nanostructured surfaces on hard and brittle materials without cracks is extremely difficult, we proposed a template-free, eco-efficient, one-step forming fabrication of hierarchical-structures on hard and brittle materials (Si) to bionic manufacture bamboo leaf surface. The obtained BLHS surfaces are not only similar to bamboo leaves in morphology but also in wetting properties. We use the bamboo-leaf-like hierarchical-structures to transform the wetting property of Si from hydrophilic to hydrophobic without coatings. Such an ability to independently tune the properties of Si surfaces without disrupting their wetting response could have important implications in the design of conducting, lap-on-chip and heat and mass transfer fields.

This research demonstrates hydrophilic wetting surface can be converted to hydrophobic surface without the need of post treatment just through eco-friendly and cost-effective approach: microstructuring surface by microgrinding.

  1. In Fig 1c the Abbott–Firestone curve of the morphology, please insert the titles for all axes and link the curve to the proper Y axe (or change the colour of Abbott–Firestone curve in red). Also, for axes in fig 5. Explain in text the complete meaning of figures. For example, the first graph displayed in figure 5 as a Histogram, gives the probability (frequency) for points to be at a given height (depth). The second graph Abbott-Firestone curve, is the cumulative curve of the distribution, displays the material ratio in function of the depth. The material ratio is the percentage of the surface above a given depth.

Response: The titles for all axes have been added in Figure 1c and Figure 5 and the colour of Abbott–Firestone curve has been changed to red in Figure 1c.

The meaning of Abbott–Firestone curve figure has been added in the revised manuscript as “The histogram in the figure gives the proportion for a given depth of microstructure. The curve marked in red color displays the microstructured surface area ratio in function of depth.” (Page 6)

  1. Describe all procedures by which the surface profile data were obtained, analyzed (software ……). Please, provide complete information about the fractal description used in the study. Each term in the presented equations must be explained.

Response: The three-dimensional (3D) topographies of the micro-machined BLHS surfaces are measured by a white light interferometer (WLI: BMT SMS Expert 3D), the Abbott–Firestone curve parameters are computed using a Taly Map Platinum measurement software available as a part of BMT SMS Expert 3D, and the two-dimensional (2D) topographies by scanning electron microscopy (SEM: TESCAN MIRA4). The acquisition and processing software of surface profile data has been added in the revised manuscript in Part 3. Materials and measurement.

Each term in the presented equations in the manuscript has been explained. The complete information about the calculating process of fractal dimension has been added in supplementary materials.

  1. Specify, the specie of investigated bamboo. Briefly describe the methodology follow in order to minimize dehydration during transportation to nearby laboratory facilities (preserved in water?) How were performed SEM analysis (on dried/wet pieces of leaves)?

Response: The specie of investigated bamboo is the Pink single bamboo (also called Bambusa chungii McClure). It was collected from Guangzhou, Guangdong, China.

The SEM photos of fresh bamboo leaves (wet pieces of leaves) are measured by environmental scanning electron microscope (ESEM, FEI QUANTA250).

These parts have been added in Part3. Materials and measurement.

  1. At presented magnification of leaf surface (figure 1b), some microstructure projections covering all the surfaces (micropapillae) and wax bumps (at nano-scales) are usually observed. Can author comment this aspect?

Response: The wetting property of a surface is a combination of surface topography and surface chemical component (Hydrophilic or hydrophobic groups, low or high surface energy materials, etc.). The wax is a low surface energy material, if the microstructured surface is covered with wax, it will be hydrophobic with low surface adhesion, as shown in the back side of the bamboo leaf. As the front side of the bamboo leaf shows a high adhesive force, we suppose that the wax isn’t a dominating factor that influences the wetting of the front side of the bamboo leaf.

  1. It is known that the surrounding viscous medium or the size of the surfaces relative to the droplets could have influenced the wetting of the droplet by introducing edge effects. The proposed model can be used as such in the case of other substrates or viscous media or does the present study need to be accommodate?

Response: The liquid viscosity influences the droplet contact angle from the liquid surface tension aspect according to Young’s equation. So, if using another liquid to measure the surface contact angle, the intrinsic contact angle θe is also different from the intrinsic contact angle of water. According to the proposed model “cosθ=(Dβ-D)rscosθe, the calculated contact angle θ will change as well.

The fractal wetting model is proposed ignoring the gravitation of droplets, so the adopted volume of the droplet is relatively small, usually at the microliter level. As for the quantitative analysis of the influences of liquid size and character on contact angles, it is in our future research work.

  1. Insert the correlation factor between preparation technologies and D. It was noted that the trend of D value of the fracture surfaces was lower when Sa was high, the dependence is almost linearly (R2~0.87).

Response: The fractal dimension D is a measure of complexity and self-similarity of surface topography. The solid surface roughness Sa is the arithmetic average deviation of the sampling area. Although the fractal dimension D and solid surface roughness Sa has a specific correlation in mathematics expression, the physics meaning is limited. Thus, we don’t think adding the correlation factor in this manuscript is necessary. But, it will be studied systematically in our further research about the influence of micromachining parameters on the forming quality of the micro-structuring surfaces.

Figure 13c shows the relationship between the measured contact angles and the fractal wetting pattern. The author writes: "The green area represents the contact angles measured on the front side of natural bamboo leaves in different growth phases" Explain what contact angles mean in different growth phases.

Response: In order to ensure data accuracy, we took the bamboo leaves in the seedling stage, long leaf stage, maturity stage, and wilting stage, respectively, and measured their contact angles.

To avoid misunderstanding, the sentence "The green area represents the contact angles measured on the front side of natural bamboo leaves in different growth phases" has been revised to “The green area in the figure is the general range of the measured contact angles on the front side of bamboo surfaces.”

  1. Insert in conclusions also a short sentences with the results obtained for solid surface roughnesses and fractal dimensions of the sample surfaces by micromachining with different diamond grindings. Please include specific and quantitative summary results in your Conclusions suitable for a broad audience.

Response: The conclusions have been revised.

  1. Revised the English.

Response: The grammar and word spelling throughout the manuscript have been revised.

For example contact angles instead of contact angels.

Response: Sorry about this mistake, the manuscript has been thoroughly revised in terms of grammar and spelling.

Reviewer 2 Report

In their paper "The transitional wettability on baboo-leaf-like ...", Jianhua Xiang and coworkers present an experimental study on diamond-ground Si surfaces.

The preparation and analysis of the ground samples seems appropriate and sound. The analysis of wettability to prove the transition from hydrophobic to hydrophilic is for sure of interest for applications as briefly mentioned by the authors and of interest for the readers of nanomaterials.

However, before the paper is ready for publication, the authors should address a number of points:

1. page 3, 2nd paragraph: "... a typical hydrophilic and hydrophobic (Fig. 1d-1e)." Please recheck this sentence.

2. Figure 2: part (c) is not described in sufficient detail in the text.

3. page 8, equation (2): what is C?

4. Figure 6: the dimensions of the inserts are missing.

5. Figure 8: The scale bars of the microscope images are barely readable.

6. page 10, last paragraph: a description on how the surface is covered with PFPE is missing.

7. page 11, 2nd paragraph: Are the given digits (40.8%, 66.6%) significant? What are the error bars?

8. Figure 12: The figure is not described in sufficient detail in the text.

9. page 12: Maybe I overlooked that, but I cannot find a proper definition of the two roughness factors \gamma_w and \gamma_c.

10. page 13, 1st paragraph: what is the third reason why the contact angle is larger as predicted by the Wenzel model and smaller as predicted by the Cassie-Baxter model?

11. references: The "[J]" should be removed from almost all references.

12. references: some references have the DOI included, some have not.

I recommend that the paper should be reconsidered after major revision.

Author Response

  1. page 3, 2nd paragraph: "... a typical hydrophilic and hydrophobic (Fig. 1d-1e)." Please recheck this sentence.

Response: The sentence has been revised to “The surfaces exhibit the typical hydrophilic and hydrophobic properties, respectively”.

  1. Figure 2: part (c) is not described in sufficient detail in the text.

Response: Figure 2c has been revised and added some details. Its description is added in the revised manuscript as: Given a grinding wheel speed N and feed speed vf, the diamond grains gradually cut into and cut out the Si surface along the cycloid track (Fig. 2c). When micro-cutting is performed in elastic-plastic removal region, the microgrooves are gradually patterned by successively replicating diamond grain profiles on the workpiece surface.

  1. page 8, equation (2): what is C?

Response: Where C is s the intercept of the double log plot of Eq. (1), it is usually a constant. The complete information about the calculating process of fractal dimension has been added in the supplementary materials.

  1. Figure 6: the dimensions of the inserts are missing.

Response: The dimensions of the inserts in Figure 6 have been added.

  1. Figure 8: The scale bars of the microscope images are barely readable.

Response: Figure 8 has been revised.

  1. page 10, last paragraph: a description on how the surface is covered with PFPE is missing.

Response: The modification of the BLHS surface is realized through coating with photocurable perfluoropolyether (PFPE), a bifunctional PFPE–urethane methacrylate, and subsequent photocuring by UV illumination. The description of how the surface is covered with PFPE has been added in the revised manuscript.

  1. page 11, 2nd paragraph: Are the given digits (40.8%, 66.6%) significant? What are the error bars?

Response: To test the BLHS surface adhesive ability, we conducted droplet stretching and compressing experiments. The tensile ratio is about 40.8% and the compression ratio is about 66.6%. The values of the results are stable and the fluctuation range is within 2.0%.

  1. Figure 12: The figure is not described in sufficient detail in the text.

Response: Figure 12 has been revised and the description of the figure has been revised to be clearer.

  1. page 12: Maybe I overlooked that, but I cannot find a proper definition of the two roughness factors \gamma_w and \gamma_c.

Response: According to Wenzel’ model, the rw is defined as the ratio of actual contact area to apparent area, as for the rc, it is the fraction of the droplet’s base area contacting with the solid according to Cassie’s model. The meanings of the parameters of rw and rc have been added in the revised manuscript.

  1. page 13, 1st paragraph: what is the third reason why the contact angle is larger as predicted by the Wenzel model and smaller as predicted by the Cassie-Baxter model?

Response: The third reason is “there is a stable intermediate state, as depicted in Fig. 8c, the liquid sinks in the microstructure but the air-chamber patterns among the “sinking” liquid and nanostructures”.

  1. references: The "[J]" should be removed from almost all references.

Response: The "[J]" has been removed from all references.

  1. references: some references have the DOI included, some have not.

Response: The references have been checked and revised.

I recommend that the paper should be reconsidered after major revision.

Response: Thanks for the reviewer’s careful and professional review. We have revised the manuscript according to the comments.

Round 2

Reviewer 1 Report

Accept in present form

Reviewer 2 Report

The authors responded satisfactorily and comprehensively to all my comments from the first expert report. So I think the paper is now ready for publication in Nanomaterials.